# Application of a Hybrid Artificial Neural Network-Particle Swarm Optimization (ANN-PSO) Model in Behavior Prediction of Channel Shear Connectors Embedded in Normal and High-Strength Concrete

**Mahdi Shariati [1,2], Mohammad Saeed Mafipour [3], Peyman Mehrabi [4], Alireza Bahadori [3], Yousef Zandi [5], Musab N A Salih [6], Hoang Nguyen [7,*], Jie Dou [8,*], Xuan Song [9] and Shek Poi-Ngian [10]**

1. Division of Computational Mathematics and Engineering, Institute for Computational Science, Ton Duc Thang University, Ho Chi Minh 758307, Vietnam; shariati@tdtu.edu.vn
2. Faculty of Civil Engineering, Ton Duc Thang University, Ho Chi Minh City 758307, Vietnam
3. School of Civil Engineering, College of Engineering, University of Tehran, Tehran 1417466191, Iran; m.saeed.mafipour@ut.ac.ir (M.S.M.); en.ar.bahadori@ut.ac.ir (A.B.)
4. MSc graduate, Department of Civil Engineering, K.N. Toosi University of Technology, Tehran 15875-4416, Iran; peyman804m@gmail.com
5. Department of Civil Engineering, Tabriz Branch, Islamic Azad University, Tabriz 5157944533, Iran; zandi@iaut.ac.ir
6. School of Civil Engineering, Faculty of Engineering, Universiti Teknologi Malaysia, Johor Bahru 81310, Malaysia; nasmusab2@live.utm.my
7. Institute of Research and Development, Duy Tan University, Da Nang 550000, Vietnam
8. Civil and Environmental Engineering, Nagaoka University of Technology, 1603-1, Kami-Tomioka, Nagaoka, Niigata 940-2188, Japan
9. Center for Spatial Information Science, the University of Tokyo, 5-1-5, Kashiwa 277-8568, Japan; songxuan@csis.u-tokyo.ac.jp
10. Construction Research Center (CRC), Institute for Smart Infrastructure & Innovative Construction (ISIIC), School of Civil Engineering, Universiti Teknologi Malaysia, Johor Bahru 81310, Malaysia; shekpoingian@utm.my
* Correspondence: nguyenhoang23@duytan.edu.vn (H.N.); douj888@vos.nagaokaut.ac.jp (J.D.)

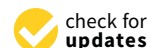

**Featured Application:** **Behavior prediction of channel shear connectors in normal and high-strength concrete (HSC) without conducting costly experiments.**

**Abstract:** Channel shear connectors are known as an appropriate alternative for common shear connectors due to having a lower manufacturing cost and an easier installation process. The behavior of channel connectors is generally determined through conducting experiments. However, these experiments are not only costly but also time-consuming. Moreover, the impact of other parameters cannot be easily seen in the behavior of the connectors. This paper aims to investigate the application of a hybrid artificial neural network–particle swarm optimization (ANN-PSO) model in the behavior prediction of channel connectors embedded in normal and high-strength concrete (HSC). To generate the required data, an experimental project was conducted. Dimensions of the channel connectors and the compressive strength of concrete were adopted as the inputs of the model, and load and slip were predicted as the outputs. To evaluate the ANN-PSO model, an ANN model was also developed and tuned by a backpropagation (BP) learning algorithm. The results of the paper revealed that an ANN model could properly predict the behavior of channel connectors and eliminate the need for

conducting costly experiments to some extent. In addition, in this case, the ANN-PSO model showed better performance than the ANN-BP model by resulting in superior performance indices.

**Keywords:** artificial neural network (ANN); particle swarm optimization (PSO); hybrid ANN-PSO; channel shear connector; high-strength concrete (HSC); behavior prediction

## 1. Introduction

Composite systems have always been of interest as they benefit from the combined properties of different materials simultaneously. Shear connectors are mainly used in steel–concrete composite systems to establish a connection through which the developed shear forces at the interface of the materials can be collected and transferred [1,2]. In composite beams with partial interaction, a specific number of shear connectors are employed along the length of beams, and these connectors primarily control the behavior of the beams under different loading conditions [3,4]. In addition, the load-bearing capacity, stiffness, and ductility of connectors highly affect the applicable theories in order for analyzing the floor systems [5–7]. In steel–concrete composite columns, shear connectors have the principal task in unifying the materials and converting them to a single unit [8]. Hence, the proper performance of composite systems largely depends on the behavior of shear connectors.

### 1.1. Channel Shear Connector

Channel shear connectors are a popular type of connectors that have been increasingly used in different countries due to three main reasons. First of all, this type of connector is more affordable than other types of connectors such as perfobond connectors, because it does not require any peculiar manufacturing process [9]. Secondly, a channel shear connector can be easily installed by even the simplest construction equipment, and it does not need any specific tool, such as a high-power generator in the case of headed stud connectors, for installation [10]. Thirdly, a semi-skilled operator is even capable of installing this connector properly, thus reducing extra costs.

In spite of the considerable advantageous of channel shear connectors, few studies have been carried out to evaluate the performance of these connectors. The load–slip behavior of channel connectors was firstly investigated by Slutter, Driscoll, and Viest et al. [11,12]. Based on these researches, provisions such as the American Institute of Steel Construction (AISC) [13] also proposed relations to calculate the shear capacity of channel connectors. The behavior of channel connectors in engineered cementitious composite (ECC), reinforced concrete (RC), and fiber-reinforced concrete (FRC) were investigated by Maleki et al. [14–16]. Shariati et al. evaluated the performance of channel connectors in lightweight aggregate concretes [17,18] and observed reductions in the shear capacity of channel connectors due to the lower strength of the concrete. Pashan and Hosain [19,20] reported the bearing capacity of channel connectors in solid slabs and metal deck slabs and proposed two mathematical relations to estimate the shear capacity of these connectors.

### 1.2. High-Strength Concrete (HSC)

The economical and engineering advantages of high-strength concrete (HSC) make it distinct from other types of concrete [21]. Used in slender structures, HSC increases the stability and bearing capacity of the structure [22,23]. In high-rise buildings, not only does it prevent unacceptable oversizing of the columns on the lower floors, it also makes it possible to construct columns with longer spans [24]. Used in bridges, it decreases the dead load of bridges by reducing the size of structural members, while prolonging the serviceable life of the bridge [25]. Moreover, not only does HSC provide a valuable alternative to engineers in challenging constructional projects, but it also enhances the strength per unit cost, unit weight, and unit volume [26]. Aside from these, HSC possesses uniform high density and very low impermeability, which make it resistant to aggressive environments [27]. However, to

benefit from the favorable advantages of HSC, its possible impacts on other structural components must be taken into account prior to utilization [28,29].

Shear connectors are one of these components that are in direct contact with HSC. The behavior of connectors significantly depends on concrete such that every type of concrete can result in a new load–slip diagram [30] and subsequently stiffness [31] and ductility [32]. Hence, having a precise estimation of shear connectors' behavior in HSC seems necessary.

### 1.3. Artificial Intelligence (AI) Techniques and Metaheuristic Algorithms

There are different available techniques for data validations and predictions such as employing artificial neural networks [33–39], the finite element method (FEM), and the finite strip method (FSM) [40,41]. FEM is a well-known method that has been developed by ABAQUS and ANSYS programs and conducted on a variety of experimental studies to either validate or predict the structural behavior of the specific specimens [42]. Although FEM has been widely used for prediction, it still has shortcomings such as considerable execution time, variation of outputs, and a large amount of data requisition [43,44]. Artificial intelligence (AI) techniques are recently playing a major role in advancing engineering goals [45–50]. Artificial neural networks (ANNs), as a sub-branch of AI techniques, are capable of solving three different types of problems comprising: (1) function approximation, (2) classification, and (3) time-series prediction [51–54]. In all of the cases, a raw model of ANN is generally developed and trained by optimization techniques. Classic algorithms such as backpropagation algorithms have been basically suggested to train ANNs [55]. However, disadvantages such as being stuck in local extremums and having trouble in crossing plateaus of the error function landscape are the deficiencies of the classic algorithms [56,57]. To address these drawbacks, metaheuristic optimization algorithms such as genetic algorithm (GA) [58], particle swarm optimization (PSO) [59], and imperialist competitive algorithm [36,60] can be used. The global search feature of these algorithms can result in improving the performance of ANN in some cases.

Over the past years, several researchers have used artificial neural networks (ANNs) and optimization techniques to solve nonlinear and sophisticated engineering problems. Ahmadi, Naderpour [61] predicted the compressive strength of concrete-filled steel tubes (CCFT) using ANN and compared their results with those of the experimental results. Reliable predictions were obtained from ANN, and it was suggested as an efficient tool in the compressive strength prediction of CCFT. The ductility of reinforced concrete (RC) beams was estimated by Bengar, Abdollahtabar [62]. They concluded that ANN models could culminate in predictions that are less scattered than statistical methods. Fedutenko, Nghiem [63] and Amirian, Dejam [64], Amirian, Fedutenko [65] investigated the application of ANN in the modeling of compaction–dilation data and evaluating the performance of unconventional oil reservoirs. Satisfactory results and the great potential of ANNs for the performance evaluation of oil reservoirs were also declared throughout these investigations. Mohammadhassani, Nezamabadi-Pour [51] studied strain in the tie section of high-strength self-compacting (HSSCC) deep beams by developing an ANN model and linear regression (LR) model. In this study, the ANN model could reach performance indices that were 90 times better than those of the LR model. Liu and Li [66] employed an ANN model for the rapid numerical simulation and seismic performance prediction of RC columns. Excellent agreement between the model and test results validated the application of the ANN model in this area, too. To assess the applicability of ANN in compressive strength prediction by P-, S-, and R-wave velocities, Park, Yoon [67] conducted an investigation and concluded that considering the P-, S-, and R-wave velocities together can lead to more accurate results than only the P-wave velocity. Chen, Fu [68] utilized a hybrid ANN-PSO model in the prediction of shear strength of squat RC walls. A comparison of this model with other predictive models revealed the superior performance of the ANN-PSO. In another interesting research, Chen, Asteris [69] employed hybrid ANN-GA and ANN-ICA models to control and secure retaining walls in dynamic conditions. In this case, the ANN-ICA model could reach better performance indices than those of ANN-GA. Koopialipoor, Jahed Armaghani [70] studied the utilization of different hybrid models including

ANN-PSO, ANN-GA, ANN-ICA, and ANN-ABC (artificial bee colony) in the prediction of slope stability under static and dynamic conditions. The results of this investigation showed the higher capability of the ANN-PSO model. In another study, Koopialipoor, Fallah [71] estimated flyrock distance by ANN-PSO, ANN-GA, and ANN-ICA. This investigation also demonstrated the finer performance of ANN-PSO in the prediction of targets.

### 1.4. Main Objectives and Scoop

The main objective of the current paper is to use artificial intelligence (AI) techniques in the load–slip behavior prediction of channel shear connectors embedded in normal and high-strength concrete (HSC). For this purpose, an artificial neural network (ANN) is developed whose weights and biases are determined by the use of particle swarm optimization (PSO) technique. Along with the ANN-PSO model, another ANN model is also developed and tuned by a conventional backpropagation (BP) algorithm. To generate the required data for the ANN-based models, several push-out tests were conducted. Channel connectors with different dimensions were embedded into normal strength concrete and HSC with two different compressive strengths. Consequently, a dataset containing 1010 data points was gathered. In addition, the modes of failure and load–slip diagrams of the channel connectors in normal concrete and HSC have been determined and discussed. Finally, the load–slip behavior of channel connectors is predicted by both of the models, and the performance of the hybrid ANN-PSO model is compared with that of the ANN-BP model.

## 2. Experimental Program

### Details of the Specimens and Test Set-Up

Eight push-out specimens were prepared for testing. These specimens were divided into two series based on concrete strength. The first series was normal strength concrete specimens, and the second one was high-strength concrete (HSC) specimens. Based on the ACI 363 [72], the compressive strength of HSC varies between 41 and 82 MPa. In this study, the compressive strength of normal specimens was 38.2 MPa, while two compressive strengths of 63 MPa and 82 MPa were adopted for the HSC specimens. Air-dry condition aggregates were used in HSC mixes. Fine aggregate was graded silica sands with a maximum nominal size of 4.75 mm, and coarse aggregate was crushed granite with a maximum nominal size of 10 mm. The used cement in all the mixes was type II Ordinary Portland Cement [73] according to ASTM C150 [74]. To attain the acceptable workability, superplasticizer (SP) was added in the mixes [75]. The SP was Rheobuild 1100 with a specific gravity of 1.195, and a pH within the range of 6.0–9.0. For the normal specimens, standard sand and gravel were used, and all the other materials were the same. All of the push-out specimens were cast in a horizontal position similar to the site situations. The reliable quality of concrete for both sides of the concrete blocks was assumed as well. All the specimens were cured in water for 28 days before testing. Standard cylinders with a diameter of 150 mm and length of 300 mm and standard cubes with a length of 100 mm were cast together with the push-out specimens to measure the compressive strength of the concrete blocks. All cylinders and cubes were cured in water until the day of testing. The concrete strength was achieved from the cylinder and the cube compression tests together. The requirements of the ASTM C39 [76] were used for the compressive strength test procedure, and the mean values of the compressive strengths were used in the calculations. Table 1 illustrates the mixture proportion of concrete materials in which Hh and Hl represent HSC specimens with higher (82 MPa) and lower (63 MPa) strengths, respectively, and N refers to normal strength concrete specimens in which ordinary materials were used.

**Table 1.** Mixture proportioning of the concrete materials.

| Mix No. | Cement (kg/m$^3$) | CA (kg/m$^3$) | FA (kg/m$^3$) | SF (kg/m$^3$) | Water (kg/m$^3$) | W/C | SP (%) | $E_c$ (GPa) | $f'_c$ (MPa) |
|---|---|---|---|---|---|---|---|---|---|
| Hh | 460 | 910 | 825 | 40 | 168 | 0.37 | 0.5 | 39 | 82 |
| Hl | 360 | 940 | 870 | - | 180 | 0.5 | 1 | 32 | 63 |
| N | 350 | 750 | 1100 | - | 133 | 0.38 | 1 | 19 | 38.2 |

CA = Coarse aggregate; FA = Fine aggregate; SF = Silica Fume; W = Water; C = Cement; SP = Superplasticizer; $E_c$ = Modulus of elasticity; $f'_c$ = Compressive strength.

Four types of channel shear connectors with the dimensions of 100 and 75 mm in height and 30 and 50 mm in length were used. The 100 mm height channels had a flange thickness of 8.5 mm and a web thickness of 6 mm, while the 75 mm channels had a flange thickness of 7.5 mm and a web thickness of 5 mm. The geometric properties of the channel connectors are presented in Table 2. In this table, the first letters of the specimens' name imply the type of concrete as mentioned before; the other numbers following the first letters also represent the height, length, web thickness, and flange thickness of the channel shear connectors, respectively.

**Table 2.** Push-out test specimens.

| Specimen | Height (mm) | Length (mm) | Web Thickness (mm) | Flange Thickness (mm) |
|---|---|---|---|---|
| Hh-100-50-6-8.5 | 100 | 50 | 6 | 8.5 |
| Hh-100-30-6-8.5 | 100 | 30 | 6 | 8.5 |
| Hl-75-50-5-7.5 | 75 | 50 | 5 | 7.5 |
| Hl-75-30-5-7.5 | 75 | 30 | 5 | 7.5 |
| N100-50-6-8.5 | 100 | 50 | 6 | 8.5 |
| N-100-30-6-8.5 | 100 | 30 | 6 | 8.5 |
| N-75-50-5-7.5 | 75 | 50 | 5 | 7.5 |
| N-75-30-5-7.5 | 75 | 30 | 5 | 7.5 |

Push-out specimens consisted of a steel I-beam with two slabs attached to each flange of the beam. One channel was welded to each flange of the beam, and two layers of steel bars with four 10 mm diameter steel bar hoops were applied in two perpendicular directions in all the slabs. Details of the test set-up can be seen in Figure 1.

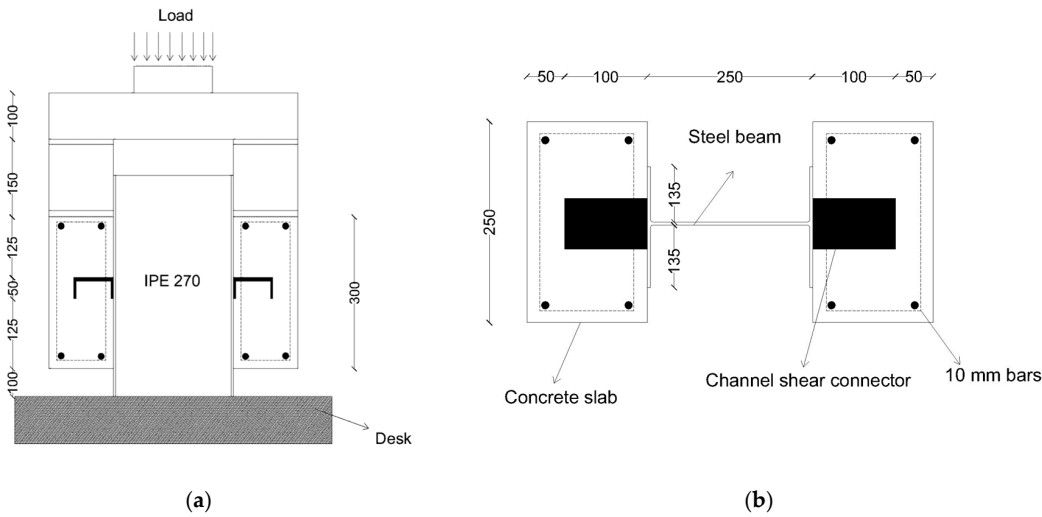

**Figure 1.** Set-up of the push-out test: (**a**) Side view; (**b**) Top view.

To determine the load–slip behavior of channel shear connectors in normal and high strength concrete, all the described specimens in Table 2 were subjected to monotonic loading. The load was applied using a 600 kN capacity universal testing machine, as illustrated in Figure 2. The test was conducted as displacement control with a loading rate of 0.04 mm/s. Due to the unidirectional nature of the load test frame, specimens were rearranged prior to loading. Monotonic loading involved slow increments of loading until failure. The steel I-beams were placed on the universal test machine deck, and the load was applied to the upper face of the concrete blocks. Since the channel connector orientation makes some variation in the ultimate strength of the connectors and relative stiffness [77,78], this matter was considered in the push-out tests, and the same orientation for the channels against the direction of loading was adopted. The actual applying load to the connectors and the relative slip between the I-shaped beam and the concrete block were automatically recorded at each time step by the universal test machine.

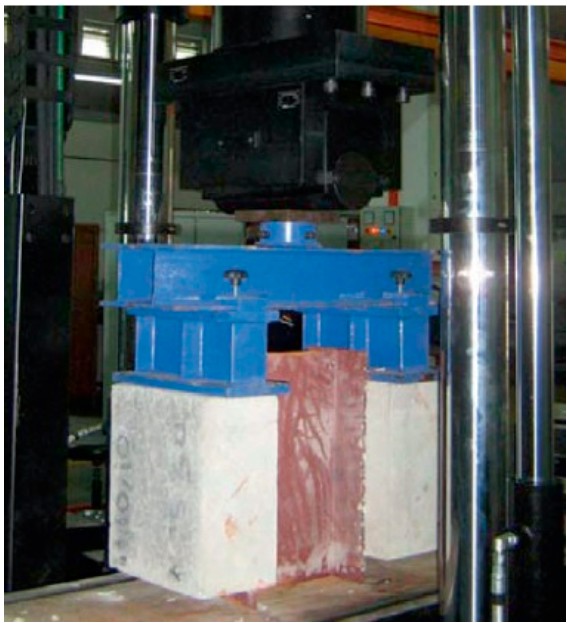

**Figure 2.** Universal testing machine [79].

## 3. Experimental Results

### 3.1. Failure Modes

The failure modes that generally occur in push-out tests can be broadly classified into two modes [10,14,18,80]. The first mode of failure is known as channel fracture with cracks in concrete blocks, and the second one is referred to as concrete crushing–splitting.

In this study, all the observed modes of failure were of the channel fracture type. Two reasons can be mentioned for this mode of failure occurring in all the specimens. The first reason can be related to the high strength of concrete specimens. In high-strength concrete (HSC), stronger molecular structures form between particles, and this in turn reduces the possibility of crack propagation. The second reason can refer to the reinforcement that was applied in concrete blocks. The reinforcement in concrete prevents cracks from propagation, and it closes the growing cracks. Hence, cracks cannot reach a critical length, which subsequently results in concrete crushing–splitting. On the other hand, microcracks, which generally remain after the welding process of channel connectors, are potential locations for crack propagation. Therefore, the mode of channel fracture is more likely to occur in concretes with higher strength or concretes with proper reinforcements.

Figure 3 shows some of the important observations at the end of the test. Figure 3a,b illustrate channel fracture in the HSC specimens and normal strength concrete, respectively. Figure 3c shows a

crack in the concrete block of the specimen N-75-50-5-7.5. Although channel fracture occurred in this specimen too, it was seen that some cracks had occurred in the concrete block. This observation also confirmed the mentioned reasons for the mode of failure, as the crack had occurred in the concrete specimen with normal strength, and reinforcement had closed the crack. Figure 3d also presents the concrete block after channel fracture.

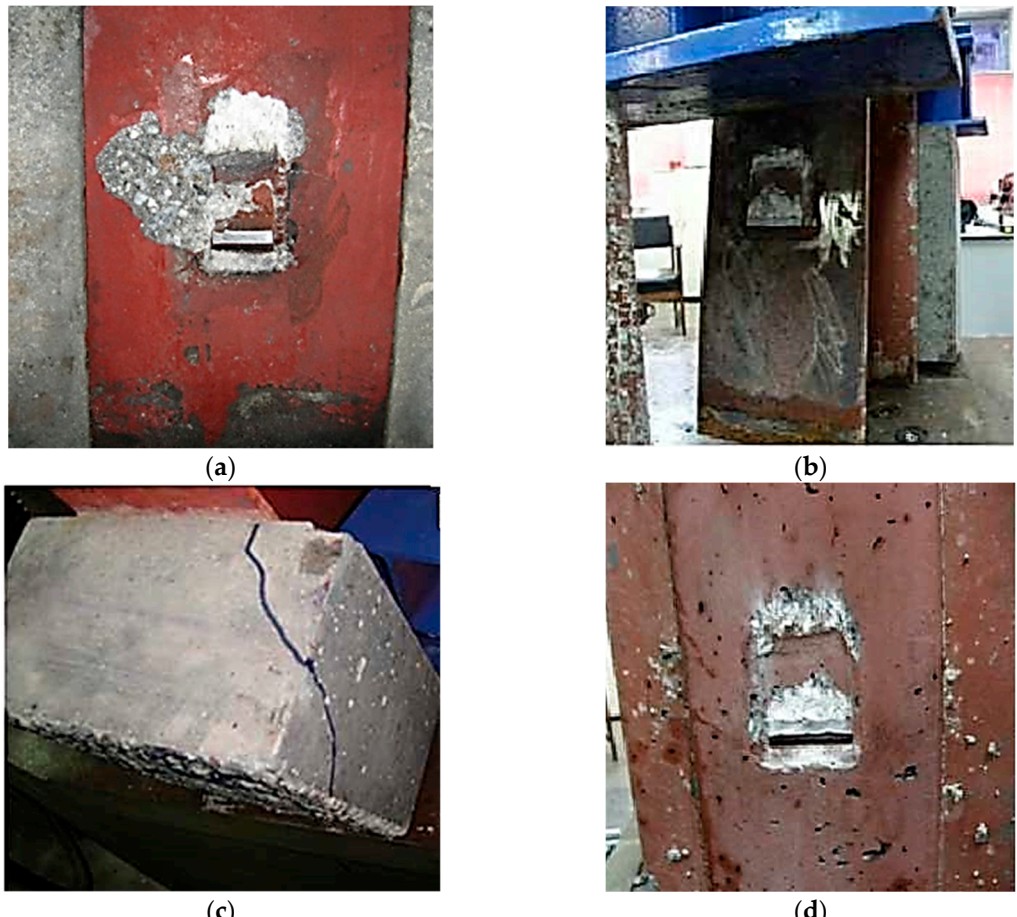

(a)          (b)

(c)          (d)

**Figure 3.** Specimens at the end of the test: (**a**) Channel fracture in high-strength concrete (HSC); (**b**) Channel fracture in normal strength concrete; (**c**) Crack in the concrete block; (**d**) Concrete block after channel fracture [79].

### 3.2. Load–Slip Behavior

The load–slip diagrams of the specimens are shown in Figure 4. In this figure, the applied load per channel versus relative displacement (slip) has been represented. As can be seen, at the peak of load, HSC specimens had been able to experience a relative slip in the rage of 10 to 14 mm, whereas this value is between 8 and 14 mm in the normal strength specimens. In this figure, Hh-100-50-6-8.5 and Hl-75-50-5-7.5 specimens have shown the highest shear capacity, and after them, the specimen of N-100-50-6-8.5 has illustrated the highest one. Therefore, it can be concluded that although the compressive strength of concrete plays a prominent role in the shear capacity of channel connectors, the length of channel connectors seems to be the most effective parameter in the shear capacity.

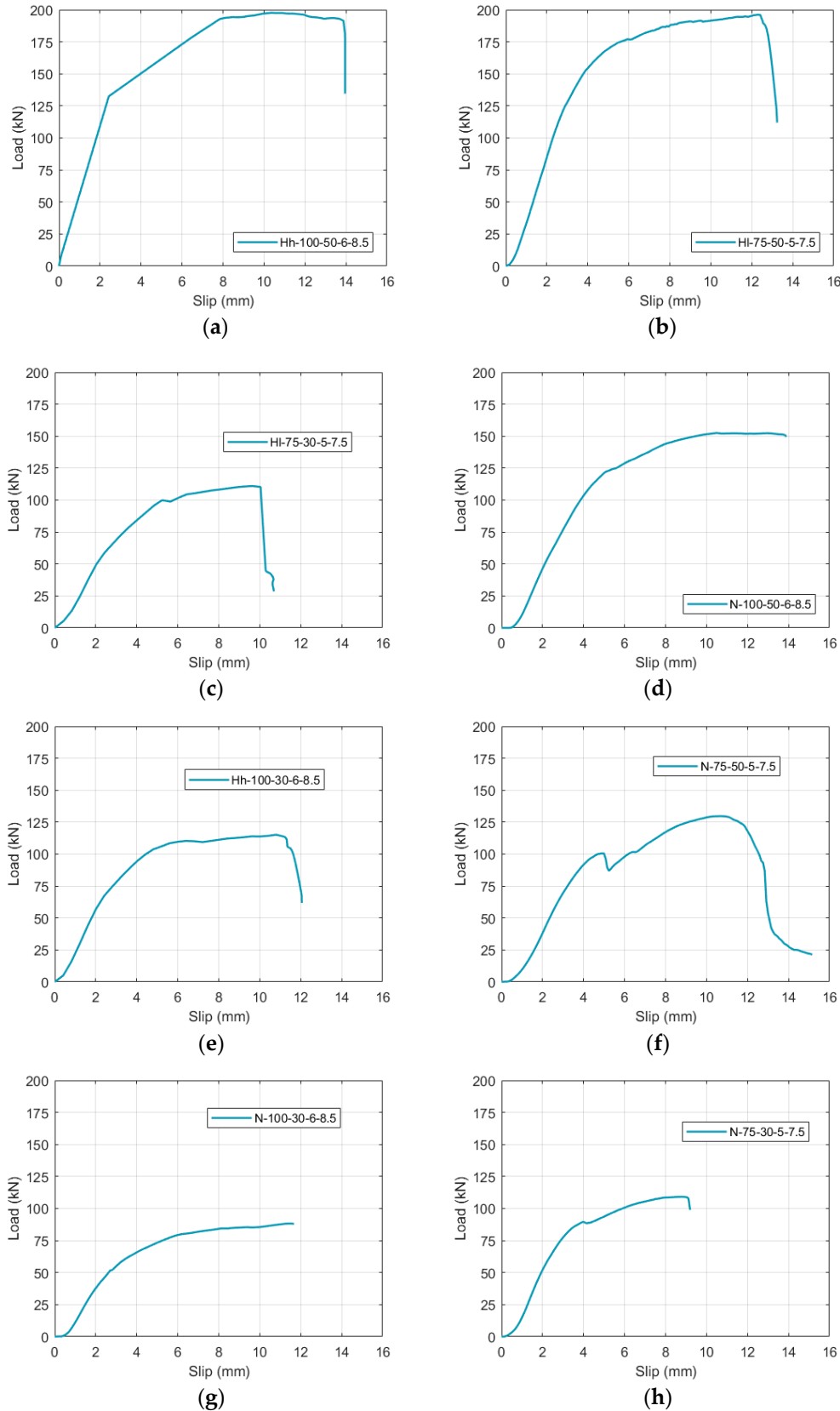

**Figure 4.** Load–slip diagrams of specimen: (**a**) Hh-100-50-6-8.5; (**b**) Hl-75-50-5-7.5; (**c**) Hl-75-30-5-7.5; (**d**) N-100-50-6-8.5; (**e**) Hh-100-30-6-8.5; (**f**) N-75-50-5-7.5; (**g**) N-100-30-6-8.5; (**h**) N-75-30-5-7.5 [80].

## 4. Methodology

### 4.1. Artificial Neural Networks (ANNs)

ANNs are intelligence tools inspired by the biological neural networks of humans and animals, which can conveniently learn patterns and predict results of a problem in high-dimensional space [81–84]. They are able to map a set of inputs to a set of outputs in a noisy and complex dataset. Multilayer perceptron (MLP) is a simple and reliable class of feed-forward ANNs. A typical MLP network contains an input layer, one or several hidden layers, and an output layer [85,86]. The input layer takes the value of inputs and sends them to the available neurons in the hidden layer. Inside each neuron, a weighted sum of inputs is calculated, and this value plus a value of bias is transformed by an activation function, as shown in Figure 5. Finally, the output signal is transferred to the neurons in the next layer.

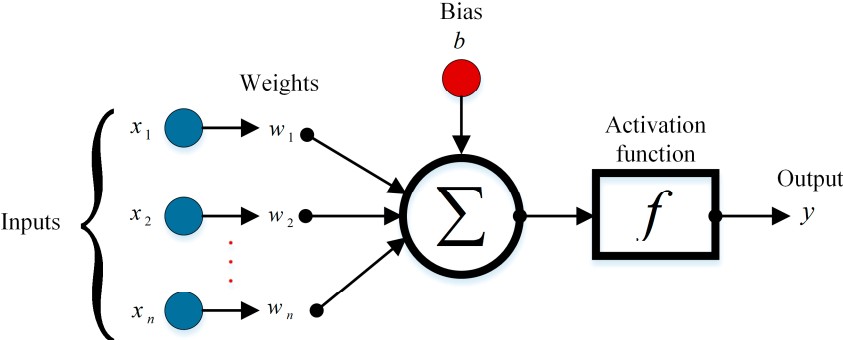

**Figure 5.** A typical neuron in an artificial neural network (ANN).

The mathematical process can be formulated as below [87]:

$$y_j = f\left(\sum_{i=1}^{n} w_{ij}x_i + b_j\right) \tag{1}$$

where $x_i$ and $y_j$ are the nodal values in the previous layer $i$ and the current layer $j$, respectively. $n$ is the total number of the nodal values received from the previous layer. $w_{ij}$ and $b_j$ are also the weights and biases of the network.

A tangent hyperbolic function has been used in this investigation, as it may lead to more accurate results [88]. This function varies between −1 and 1, and is defined as follows:

$$y_j = f(net) = \frac{2}{1 + e^{-2.net}} - 1 \tag{2}$$

where $y_j$ is the output signal, and $f$ is the activation function in the terms of the calculated network value (net).

Neural networks need to be trained to show efficient performance. Training means that the weights and biases of the network are determined such that the minimal error between targets (actual values) and outputs (network values) occurs [81]. Hence, the training process of neural networks culminates in a minimization problem. Backpropagation (BP) algorithms are commonly used in order to train neural networks [55]. The Levenberg–Marquardt algorithm (LMA) is often the fastest BP algorithm in training [89,90]; thus, LMA is applied as the BP algorithm in this study.

### 4.2. Particle Swarm Optimization (PSO)

Particle swarm optimization (PSO) is an intelligence evolutionary algorithm that was inspired by the social behavior of bird flocking or fish schooling. The PSO technique was firstly proposed by Kennedy and Eberhart in 1997 [59]. This algorithm benefits from a very fast convergence rate among the other evolutionary algorithms, and it is basically continuous [91]. Hence, it has been used

satisfactorily in many engineering problems [92–94]. In this method, a cost function that should be minimized or maximized is initially defined. Then, a swarm of particles is created and distributed in the $D$ dimensional space of the problem. Each particle contains the variables of the problem so that for each particle, the cost (fitness) function can be calculated. Finally, the velocity and the position of each particle is updated according to the following formulations until the algorithm converges [95].

$$V_i^{k+1} = w.V_i^k + c_1.r_1.\left(P_{best,i}^k - \rho_i^k\right)/\Delta t + c_2.r_2.\left(G_{best}^k - \rho_i^k\right)/\Delta t \tag{3}$$

$$\rho_i^{k+1} = \rho_i^k + V_i^{k+1}.\Delta t \tag{4}$$

where the subscripts $i$ and $k$ denote the particle and the iteration number, respectively. $\rho_i = \left\{\rho_{i1}, \rho_{i2}, \ldots, \rho_{ij}, \ldots, \rho_{iD}\right\}$ and $V_i = \left\{v_{i1}, v_{i2}, \ldots, v_{ij}, \ldots, v_{iD}\right\}$ are the position and velocity vectors, respectively. The vectors $P_{best,i}^k = \left\{p_{i1}, p_{i2}, \ldots, p_{ij}, \ldots, p_{iD}\right\}$ and $G_{best}^k = \{g_1, g_2, \ldots, g_D\}$ are the best position of the *i*th particle over its history up to iteration $k$, and the position of the best particle in the swarm in iteration $k$, respectively. $i = 1, 2, 3, \ldots, N$ is a counter to the number of particles, and $D$ is the number of problem dimensions or variables. In addition, $C_1$ is a cognitive parameter indicating the degree of local search, whereas $C_2$ is a social parameter to reflect the global search level. Besides, $r_1$ and $r_2$ are two independent random numbers uniformly distributed between 0 and 1, and $w$ is the inertial weight used to preserve the previous velocity of the particles during the optimization process. $\Delta t$ is the time interval in which the position and velocity are updated; this parameter is usually considered to be equal to 1.

*4.3. Hybrid ANN-PSO*

As mentioned previously, the training process of ANN leads to a minimization problem that can be solved by classic or metaheuristic algorithms. In a hybrid ANN-PSO model, PSO is involved to minimize the errors of the ANN by determining the optimum values for the weights and biases of the model [96]. Therefore, in this problem, variables are the weights and biases, and the feasible space of the problem depends on the interval at which these variables vary. The cost function (fitness function) of the *i*th particle can be defined in the term of root mean squared error as follows [97]:

$$E(w_i, b_i) = \sqrt{\frac{1}{S} \sum_{k=1}^{S} \left[\sum_{l=1}^{O} \{T_{kl} - P_{kl}(w_i, b_i)\}^2\right]} \tag{5}$$

where $E$ is the cost (fitness) value, $T_{kl}$ is the target value, $P_{kl}$ is the predicted output based on $w_i$ (weights) and $b_i$ (biases), $S$ is the number of training samples, and $O$ is the number of neurons.

To have a hybrid ANN-PSO model, these steps can be followed:

1. Considering a number of neurons in the hidden layer, develop a neural network with initial weights and biases.
2. Reform the weights and biases in a way where they can represent the location of a particle in the $D$ -dimensional space of the problem, where $D$ is the total number of weights and biases.
3. For each of the particles in every iteration, we can predict output values and then calculate the value of the presented cost function in Equation (5).
4. Update the location of particles by the PSO algorithm for a specific number of populations and iterations until the target is achieved (i.e., the cost function is minimized).

## 5. Models Development

*5.1. Data and Preparation*

The used data in this investigation was obtained from the load–slip diagrams of each test specimen. A dataset containing 1010 data points was totally collected. Table 3 shows the details of the input and output parameters.

**Table 3.** Inputs and outputs.

| Inputs and Outputs | Minimum | Maximum | Average |
| --- | --- | --- | --- |
| Flange thickness (mm) | 7.50 | 8.50 | 8.00 |
| Web thickness (mm) | 5.00 | 6.00 | 5.49 |
| Height (mm) | 75.00 | 100.00 | 87.50 |
| Length (mm) | 30.00 | 50.00 | 41.90 |
| Compressive strength (MPa) | 38.20 | 82.00 | 50.97 |
| Slip (mm) | 0.00 | 15.15 | 6.99 |
| Load (kN) | 0.00 | 197.63 | 105.67 |

As the problem of prediction is nonlinear and the presented activation function in Equation (2) varies between −1 and 1, it is better to normalize the data. For this purpose, preprocessing and postprocessing can be conducted [53], and the input data is normalized in the interval between −1 and 1 by the following formulas:

$$x_i = \frac{x_{io} - x_{min}}{x_{max} - x_{min}} \times 2 - 1 \tag{6}$$

$$y_i = \frac{y_{io} - y_{min}}{y_{max} - y_{min}} \times 2 - 1 \tag{7}$$

where $x_{io}$ and $x_i$ are the $i$th component of each input vector before and after normalization, respectively, and $y_{io}$ and $y_i$ are the $i$th component of the output vector before and after normalization, respectively. $x_{min}$, $x_{max}$, $y_{min}$, and $y_{max}$ are also the minimum and maximum value of each input and output vector, respectively.

*5.2. Performance Evaluation*

To evaluate the performance of the models, 70% of the data has been randomly devoted to the training phase, and the other 30% is assigned to the testing phase. Then, the root mean squared error (*RMSE*), Pearson correlation coefficient (*r*), and determination coefficient ($R^2$) are employed as performance indices of the models. These statistical indicators can be characterized as below:

$$RMSE = \sqrt{\frac{\sum\limits_{k=1}^{S} (P_k - T_k)^2}{S}} \tag{8}$$

$$r = \frac{S\left(\sum\limits_{k=1}^{S} T_k \times P_k\right) - \left(\sum\limits_{k=1}^{S} T_k\right) \times \left(\sum\limits_{k=1}^{S} P_k\right)}{\sqrt{\left(S\sum\limits_{k=1}^{S} T_k^2 - \left(\sum\limits_{k=1}^{S} T_k\right)^2\right) \times \left(S\sum\limits_{k=1}^{S} P_k^2 - \left(\sum\limits_{k=1}^{S} P_k\right)^2\right)}} \tag{9}$$

$$R^2 = \frac{\left[\sum\limits_{k=1}^{S} \left(T_k - \overline{T_k}\right) \cdot \left(P_k - \overline{P_k}\right)\right]^2}{\sum\limits_{k=1}^{S} \left(T_k - \overline{T_k}\right) \cdot \sum\limits_{k=1}^{S} \left(P_k - \overline{P_k}\right)} \tag{10}$$

where $P$ and $T$ are the predicted and target values, and $S$ is the total number of training or testing samples, respectively.

This is also important to note that all the codes were developed in the MATLAB environment.

*5.3. ANN Architecture*

The efficiency of ANN models depends on the architecture of the neural network (NN) i.e., the number of hidden layers and number of neurons. A single hidden layer architecture with the different

number of neurons in the hidden layer was adopted for the ANN models, as it generally leads to better results. Each model was run three times; then, the average values of *RMSE*, $R^2$, and *r* were determined for each ANN model. Tables 4 and 5 show the performance indices of the ANN models for load and slip outputs, respectively.

**Table 4.** Performance of the ANN model for different architectures (output = load). RMSE: root mean squared error.

| Number of Neurons | Train | | | Test | | |
|---|---|---|---|---|---|---|
| | *r* | $R^2$ | *RMSE* (kN) | *r* | $R^2$ | *RMSE* (kN) |
| 5 | 0.917 | 0.841 | 22.049 | 0.911 | 0.829 | 22.791 |
| 8 | 0.944 | 0.891 | 18.221 | 0.940 | 0.883 | 18.896 |
| 10 | 0.950 | 0.903 | 17.215 | 0.958 | 0.904 | 17.488 |
| 15 | 0.947 | 0.897 | 17.583 | 0.950 | 0.902 | 17.464 |

**Table 5.** Performance of the ANN model for different architectures (output = slip).

| Number of Neurons | Train | | | Test | | |
|---|---|---|---|---|---|---|
| | *r* | $R^2$ | *RMSE* (mm) | *r* | $R^2$ | *RMSE* (mm) |
| 5 | 0.858 | 0.737 | 2.077 | 0.824 | 0.680 | 2.359 |
| 8 | 0.874 | 0.764 | 1.983 | 0.821 | 0.675 | 2.335 |
| 10 | 0.884 | 0.781 | 1.913 | 0.831 | 0.687 | 2.394 |
| 15 | 0.882 | 0.778 | 1.915 | 0.827 | 0.685 | 2.382 |

As can be seen in these tables, selecting the number of 10 neurons can lead to better results by having the highest values of *r* and $R^2$, and the lowest *RMSE* value. Hence, an ANN with a single hidden layer containing 10 neurons was considered, as represented in Figure 6.

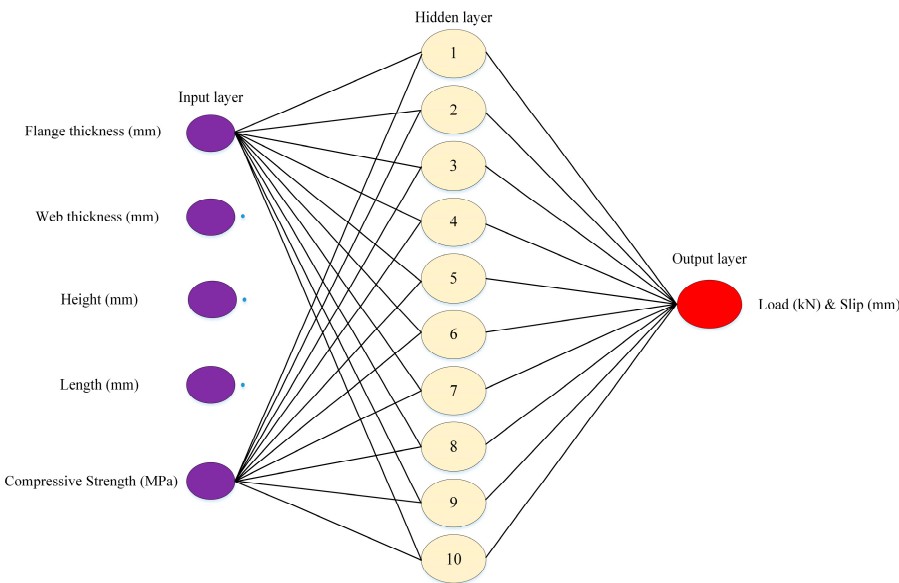

**Figure 6.** ANN architecture.

## 5.4. PSO Parameters

Parameters such as $C_1$, $C_2$, $w$, and the number of particles (population) should be also determined in the PSO algorithm. For this purpose, the ANN-PSO model was run several times for different parameters. It was seen that the fastest convergence rate and the best cost values for the PSO algorithm

could be obtained when $C_1$ and $C_2$ are equal to 1 and 2, respectively. In addition, an initial value of $w = 1$ with a damping coefficient of 0.99 has been used. To determine the number of particles (swarm size), the ANN-PSO model was rerun for different swarm sizes. A summary of the obtained values of *RMSE*, $R^2$, and *r* in the different number of particles can be seen in Tables 6 and 7.

**Table 6.** The number of particles impact on the results (output = load).

| Number of Particles | Max Iteration | Train | | | Test | | |
|---|---|---|---|---|---|---|---|
| | | *r* | $R^2$ | *RMSE* (kN) | *r* | $R^2$ | *RMSE* (kN) |
| 25 | 1000 | 0.965 | 0.931 | 14.237 | 0.968 | 0.938 | 14.030 |
| 35 | 1000 | 0.971 | 0.944 | 13.394 | 0.969 | 0.939 | 13.818 |
| 45 | 1000 | 0.972 | 0.945 | 13.074 | 0.970 | 0.941 | 13.129 |
| 65 | 1000 | 0.968 | 0.937 | 13.823 | 0.963 | 0.927 | 14.751 |
| 85 | 1000 | 0.966 | 0.933 | 14.026 | 0.967 | 0.935 | 14.441 |

**Table 7.** The number of particles impact on the results (output = slip).

| Number of Particles | Max Iteration | Train | | | Test | | |
|---|---|---|---|---|---|---|---|
| | | *r* | $R^2$ | *RMSE* (mm) | *r* | $R^2$ | *RMSE* (mm) |
| 25 | 1000 | 0.893 | 0.797 | 1.836 | 0.773 | 0.597 | 2.667 |
| 35 | 1000 | 0.887 | 0.787 | 1.882 | 0.784 | 0.615 | 2.618 |
| 45 | 1000 | 0.901 | 0.811 | 1.785 | 0.824 | 0.680 | 2.311 |
| 65 | 1000 | 0.912 | 0.831 | 1.685 | 0.791 | 0.625 | 2.538 |
| 85 | 1000 | 0.892 | 0.795 | 1.850 | 0.801 | 0.641 | 2.492 |

Figure 7 demonstrates the impact of the number of particles on the root mean squared error (*RMSE*). As can be observed in these diagrams, the lowest value of *RMSE* is obtained when the number of 45 particles is selected. More importantly, in this number of particles, the lowest difference between the *RMSE* values in the training and testing phases has been obtained. This implies that in this number of particles, overfitting is less likely to occur. Considering these facts, the number of 45 particles has been adopted in the PSO algorithm.

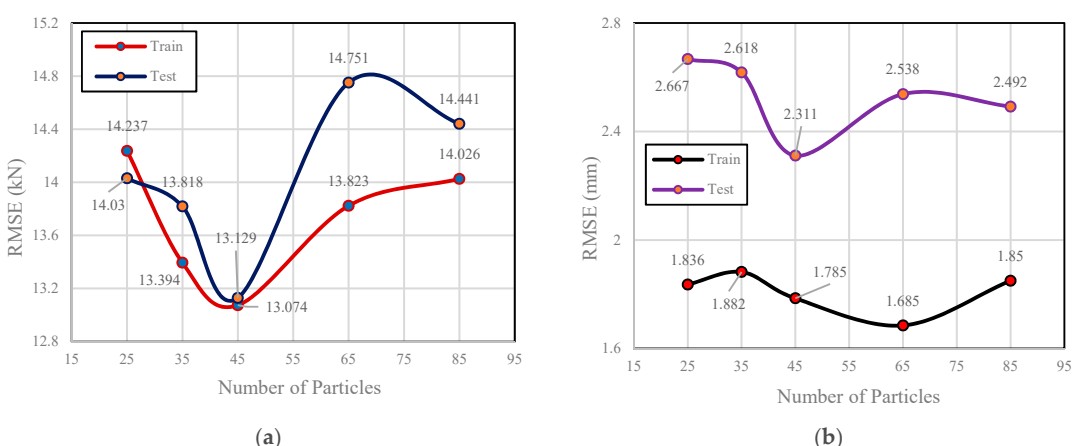

**Figure 7.** Impact of the swarm size on *RMSE* value: (**a**) Output = load; (**b**) Output = slip.

## 6. Results and Discussion

As described before, two ANN models have been considered in this study. The first model is an AAN-BP model whose weights and biases are determined by a backpropagation algorithm (BP) i.e., the Levenberg–Marquardt algorithm (LMA), and the second one is the ANN-PSO model, which is tuned by the PSO algorithm. The same architecture of 10 neurons in a single hidden layer has been

adopted in both of the models. Most (70%) of the inputs were randomly assigned to the training phase, and the remaining 30% were assigned to the testing phase.

Figure 8 illustrates the predicted and measured slip by ANN-BP and ANN-PSO models in scatter diagrams. Figure 8a shows the training phase of the ANN-BP model, whose performance parameters are $r = 0.889$, $R^2 = 0.791$, and $RMSE = 1.863$ (mm). Figure 8b depicts the testing phase of the ANN-BP model with performance parameters of $r = 0.804$, $R^2 = 0.646$, and $RMSE = 2.569$ (mm). As can be realized, the ANN-BP model has been able to show acceptable performance in the prediction of the slip, since the values of $r$ and $R^2$ are near to 1, and the $RMSE$ value is almost low in the slip range of 0 to 16 mm. Figure 8c represents the training phase of the ANN-PSO model. Performance indices of the ANN-PSO model in this phase are $r = 0.908$, $R^2 = 0.824$, and $RMSE = 1.708$ (mm). As can be seen, some improvement in the performance of the ANN has been resulted in this phase by using the PSO algorithm such that the $r$ and $R^2$ values have increased, and the $RMSE$ value has decreased. Figure 8d shows the testing phase of the ANN-PSO model with performance indices of $r = 0.865$, $R^2 = 0.750$, and $RMSE = 2.069$ (mm). The testing phase has also improved with respect to that of the ANN-BP model.

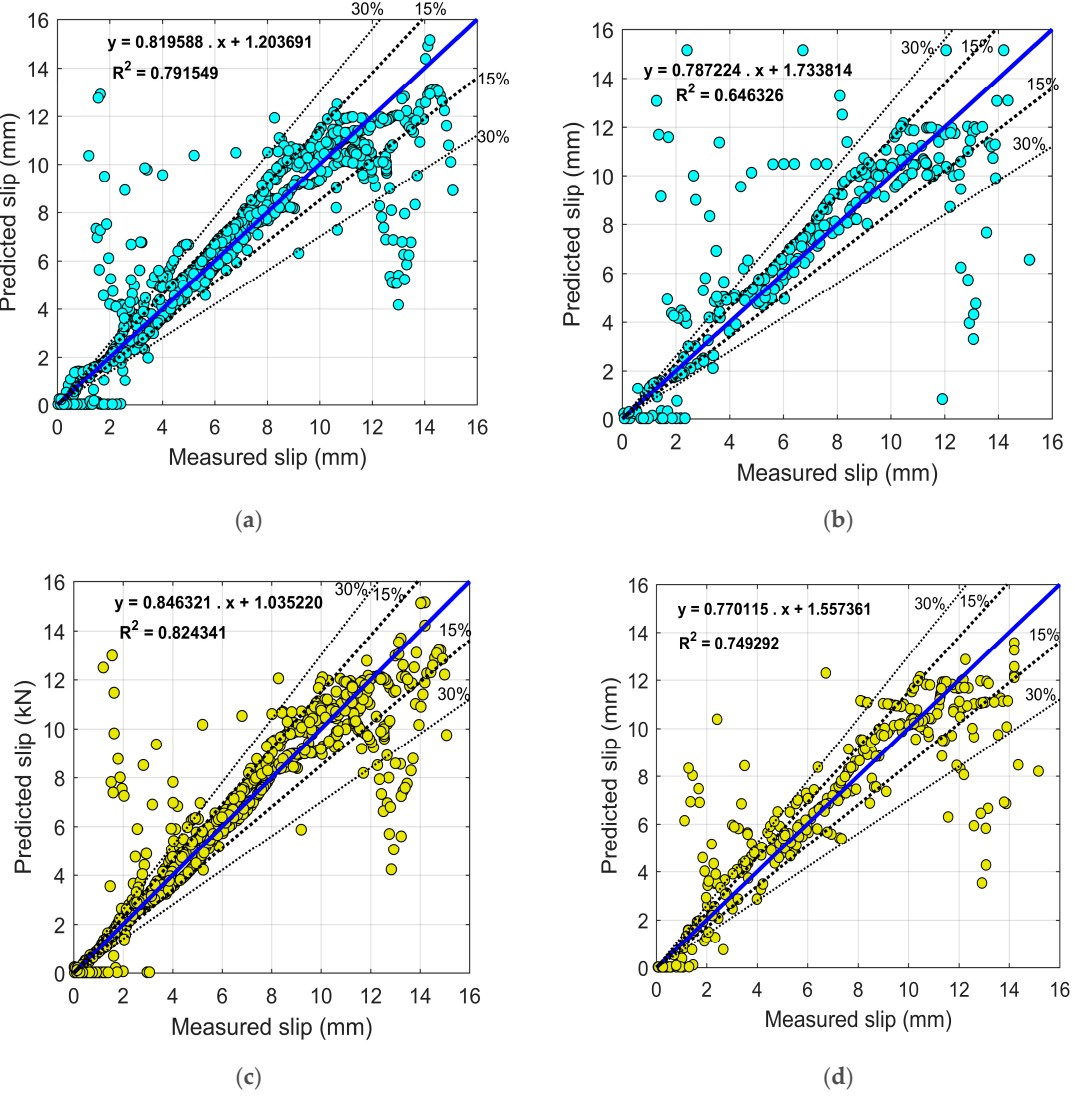

**Figure 8.** Comparison of the predicted and measured slip: (**a**) Training phase of the ANN-BP model (**b**) Testing phase of the ANN-BP model (**c**) Training phase of the ANN-PSO model (**d**) Testing phase of the ANN-PSO model. BP: backpropagation, PSO: particle swarm optimization.

Figure 9 shows the capability of the models in the testing phase to predict each of the measured values of the test samples. As can be realized, both of the models are capable of predicting most of the test samples closely. However, the better performance of the ANN-PSO model is almost clear.

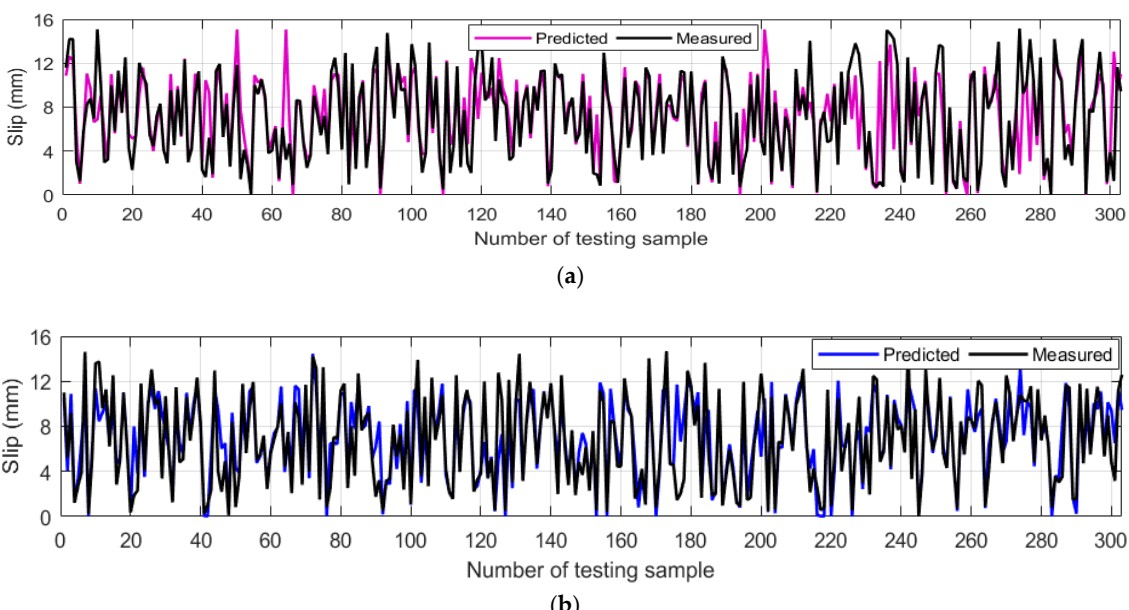

**Figure 9.** Slip prediction in the testing phase: (**a**) ANN-BP model; (**b**) ANN-PSO model.

Table 8 briefly shows the obtained performance indices from the ANN-BP and ANN-PSO models in the prediction of slip. All the performance values validate the superior performance of the ANN-PSO model.

**Table 8.** Performance indices of the ANN models in the prediction of slip.

| | Slip | | | | | |
|---|---|---|---|---|---|---|
| **Performance Indices** | **Training** | | | **Testing** | | |
| | $r$ | $R^2$ | *RMSE* (mm) | $r$ | $R^2$ | *RMSE* (mm) |
| ANN-PSO | 0.908 | 0.824 | 1.708 | 0.865 | 0.750 | 2.069 |
| ANN-BP | 0.889 | 0.791 | 1.863 | 0.804 | 0.646 | 2.569 |

Figure 10 demonstrates the results of the ANN-BP and ANN-PSO models in the prediction of the load. Figure 10a illustrates the training phase of the ANN-BP model. Performance indices of the model in this phase have been determined as $r = 0.955$, $R^2 = 0.912$, and *RMSE* = 16.747 (kN). Figure 10b represents the testing phase of the ANN-BP model with performance parameters of $r = 0.950$, $R^2 = 0.902$, and *RMSE* = 17.109 (kN). High values of $r$ and $R^2$ and a low value of *RMSE* imply the excellent performance of the ANN-BP model. Figure 10c shows the training phase of the ANN-PSO model; the performance indices are $r = 0.972$, $R^2 = 0.944$, and *RMSE* = 13.342 (kN). As can be realized, the ANN-PSO model has shown better performance in the training phase in comparison with that of the ANN-BP model. Figure 10d also demonstrates the testing phase of the ANN-PSO model with performance indices of $r = 0.970$, $R^2 = 0.940$, and *RMSE* = 13.342 (kN). The testing phase has also improved with respect to that of the ANN-BP model. Most importantly, the close values of performance indices in the training and testing phases confirm the high reliablity of the models.

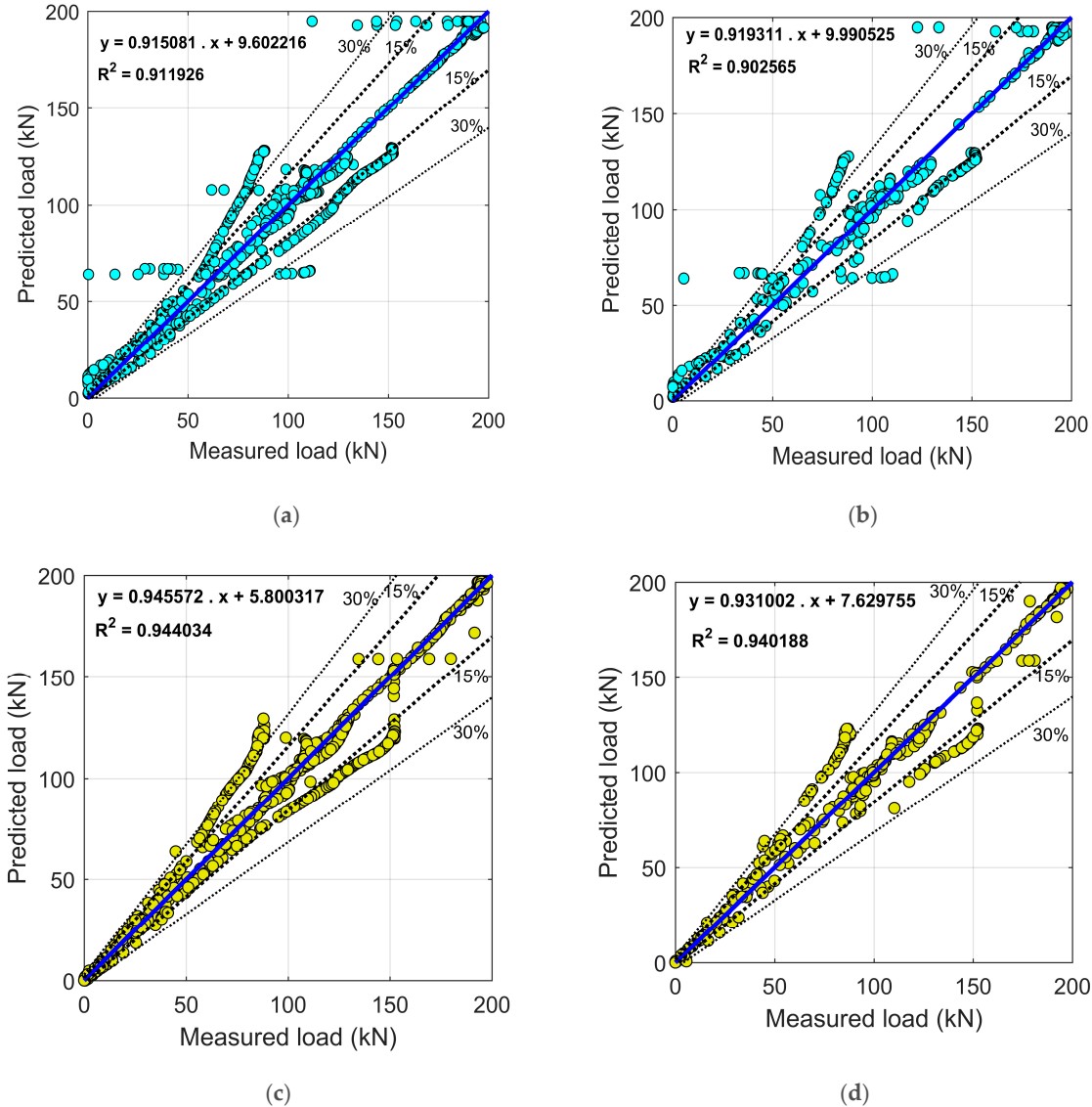

**Figure 10.** Comparison of the predicted and measured load: (**a**) Training phase of the ANN-BP model (**b**); Testing phase of the ANN-BP model; (**c**) Training phase of the ANN-PSO model; (**d**) Testing phase of the ANN-PSO.

The capability of the models in the prediction of each test sample is shown in Figure 11. Highly close prediction of the models and better performance of the ANN-PSO model is obvious in this figure.

A summary of the obtained performance values from the ANN-BP and the ANN-PSO models in the load prediction of channel shear connectors has been also illustrated in Table 9. The superior performance of the ANN-PSO model can be seen in this table.

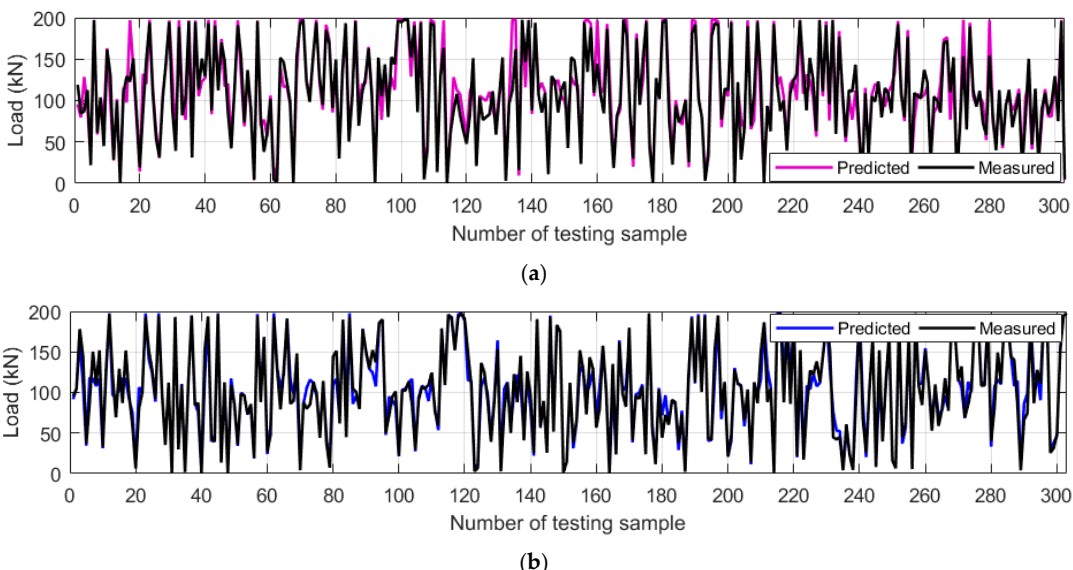

**Figure 11.** Load prediction in the testing phase: (**a**) ANN-BP model; (**b**) ANN-PSO model.

**Table 9.** Performance indices of the ANN models for the load prediction.

| | **Load** | | | | | |
| --- | --- | --- | --- | --- | --- | --- |
| **Performance Indices** | **Training** | | | **Testing** | | |
| | *r* | $R^2$ | *RMSE* (kN) | *r* | $R^2$ | *RMSE* (kN) |
| ANN-PSO | 0.972 | 0.944 | 13.342 | 0.970 | 0.940 | 13.342 |
| ANN-BP | 0.955 | 0.912 | 16.747 | 0.950 | 0.902 | 17.109 |

## 7. Conclusions

In the current study, the capability of artificial intelligence (AI) techniques in the load–slip behavior prediction of channel shear connectors was examined. Specimens of channel connectors with different dimensions were embedded in normal strength concrete and high-strength concrete (HSC). Several push-out tests were carried out, and the load–slip diagrams of the specimens were determined. In addition, the modes of failure in the specimens were reported. In order to eliminate the need for conducting the costly and time-consuming push-out tests, a hybrid artificial neural network-particle swarm optimization (ANN-PSO) model was developed. Besides, to validate and examine the performance of the ANN-PSO model, an artificial neural network (ANN) was also created and tuned by a commonly used backpropagation (BP) algorithm. Finally, the load–slip behavior of the tested channel connectors was predicted by both of the ANN-PSO and ANN-BP models. The following conclusions can be drawn from the paper:

- In all the tested specimens, the mode of channel fracture was only seen. Therefore, it can be concluded that channel connectors are more likely to have this mode of failure in HSC. In addition, reinforcement could be mentioned as another reason for occurring this mode of failure, as the normal strength specimens also experienced the same failure even though they had lower compressive strengths.

- At the peak of load, channel connectors embedded in HSC showed a higher range of slip in comparison with that of the normal strength specimens. This shows that the type of concrete and compressive strength can change the behavior and ductility of channel connectors.

- Although the compressive strength of concrete was effective in the shear capacity of channel connectors and the specimens with higher compressive strengths resulted in higher shear capacities,

it seems that the length of channel connector is the most efficient parameter as the connectors with longer lengths could meet the higher shear demands.

- It was observed that developing an ANN model would be practical in the load–slip behavior prediction of channel connectors, thus eliminating the need for conducting costly experiments to some extent.
- Although both the ANN-BP and ANN-PSO models showed acceptable performance in the load–slip prediction, it has been shown that the PSO algorithm is capable of improving the accuracy of the prediction. Therefore, this algorithm can be practically used in the case of the load–slip behavior prediction of channel shear connectors.

**Author Contributions:** Conceptualization: M.S., M.S.M., and P.M.; investigation: A.B.; experiment: M.S. and H.N.; analysis of experimental results: M.N.A.S. and M.S.M.; data preparation: S.P.-N. and P.M.; methodology: P.M. and Y.Z.; models development: P.M., M.S.M., X.S., and J.D.; resources: H.N.; writing—Original draft preparation, A.B.; writing—Review and editing: M.S.M. and P.M.; project administration: M.S.; funding acquisition: H.N.

**Funding:** This research received no external funding.

**Conflicts of Interest:** The authors declare no conflict of interest.

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
