# Peer review of "Application of a Hybrid Artificial Neural Network-Particle Swarm Optimization (ANN-PSO) Model in Behavior Prediction of Channel Shear Connectors Embedded in Normal and High-Strength Concrete"

_applsci, doi:10.3390/app9245534_

Round 1

Reviewer 1 Report

The paper discusses the design of an artificial neural network (ANN) to predict channel connectors embedded in normal and high strength concrete based on an experimental campaign.

Positive

The paper is focused on an original ANN design in the field of structural engineering and this is interesting for scientific community. The ANN design is based on an original experimental campaign through failure tests on connections. The ANN prediction is satisfactory based on results shown in Fig.8. The manuscript follows an opportune scientific approach. The paper is well written.

Negative

The paper’s main focus (i.e. shear connections) is too specific to be generalized because the ANN depend on shear connection typology. exeprimental tests are few The ANN architecture should be discussed better in Paragraph 4.3. Figure 8 should be improved.

Reviewer 2 Report

The article is sufficiently novel and interesting to warrant publication and it adheres to the journal's standards. The article is clearly laid out. All the key elements are present: abstract, introduction, methodology, results, discussion and conclusions. The title clearly describes the article and the abstract reflects the content of the article. The introduction contains a brief description of the actual state-of-the-art, and clearly state the problem being investigated. The authors accurately explain what they discovered in the research. The claims in conclusion are supported by the results. The references are accurate. The reviewer recommends accepting the paper for publication.

Reviewer 3 Report

The introduction should be improved with deeper literature review for all parts of it. For example, "The channel shear connector is a popular type of shear connectors, especially in the developing countries". Why do you choose this connector, and why do you focus on the developing countries?. 

Or "Several studies have been carried out to evaluate the performance of channel connectors." What does the meaning of "several"? Can you figure out how many?. In this part, only 10 references are mentioned. This is not "several".

"High‐strength concrete (HSC) is an interesting type of concrete that can be used in different applications". Please give more reasons why it is "interesting"?. Also, I don't believe there are a few pieces of research regarding to this matter with your review like this: "However, although several publications can be found in the literature on the performance of headed stud connectors in HSC [25, 29], few researchers have investigated the behavior of the channel connectors in this type of concrete."

Reviewer 4 Report

The paper is well developed. It is a novel topic and quite useful in research. An artificial neural network, represents a breakthrough and helps optimize resources and manufacturing costs. I invite you to continue working in this line.

Round 2

Reviewer 3 Report

The manuscript has been improved sufficiently.